Red deer synchronise their activity with close neighbours

Rands Sean A. sean.rands@bristol.ac.uk
Muir Hayley
Terry Naomi L.
School of Biological Sciences, University of Bristol , Bristol , UK
Rogers Lesley
Electronic publication date: 2014 Apr 10
Publication date: 2014
Volume: 2
Electronic Location ID: e344
Received 2013 Dec 20; Accepted 2014 Mar 25
Copyright: © 2014 Rands et al.
Copyright year: 2014
Copyright holder: Rands et al.
License: This is an open access article distributed under the terms of the Creative Commons Attribution License, which permits unrestricted use, distribution, and reproduction in any medium, provided the original author and source are credited.
License URL: https://creativecommons.org/licenses/by/3.0/

Keywords: Synchrony, Neighbours, Collective behaviour, Modelling, Social networks

Funding: This work was not funded.

==============================
Models of collective animal behaviour frequently make assumptions about the effects of neighbours on the behaviour of focal individuals, but these assumptions are rarely tested. One such set of assumptions is that the switch between active and inactive behaviour seen in herding animals is influenced by the activity of close neighbours, where neighbouring animals show a higher degree of behavioural synchrony than would be expected by chance. We tested this assumption by observing the simultaneous behaviour of paired individuals within a herd of red deer Cervus elaphus. Focal individuals were more synchronised with their two closest neighbours than with the third closest or randomly selected individuals from the herd. Our results suggest that the behaviour of individual deer is influenced by immediate neighbours. Even if we assume that there are no social relationships between individuals, this suggests that the assumptions made in models about the influence of neighbours may be appropriate.

Introduction

Many animals form groups at some point in their life cycle. In most cases, these groups occur because there is some benefit from being in the group to each of its members (Krause & Ruxton, 2002), suggesting that the behaviour of each individual must in part be both influenced by and directed towards behaving as part of the group. Models of collective behaviour (Camazine et al., 2001; Sumpter, 2010) frequently consider the behaviours of groups that emerge from the combined actions of the individuals within the group. These models are good at creating simulations of the movements and decision-making processes of groups that appear to behave in very similar ways to what is seen in nature, but very different models can produce similar phenomena. In order to identify which modelled processes are appropriate, it is essential to challenge these models with empirical data. However, the noisiness of biological systems (either from observational error or biological variation between individuals) increases the difficulty of testing whether the interaction rules used in these models are appropriate (Mann, 2011).

Many of the models and associated empirical studies that describe collective behaviour typically consider individuals that are influenced by other group members who are in close proximity, either within a physical ‘metric’ distance of a focal individual (Couzin et al., 2002; Herbert-Read et al., 2011; Rands et al., 2004; Rands et al., 2006; Romey & Vidal, 2013), or according to a topologically-defined network of interacting individuals (Bode, Franks & Wood, 2011a; Camperi et al., 2012; Nagy et al., 2010). Other influential models of movement involve changes in behavioural states, considering the departure and leadership decisions made by groups of moving animals (Fernandez & Deneubourg, 2011; Pillot et al., 2011; Sueur et al., 2011), where the behavioural state change experienced by individuals is the switch from being static to moving. Other studies of behavioural state changes have considered how local interactions govern changes between being vigilant and non-vigilant (Beauchamp, Alexander & Jovani, 2012), or being active and inactive according to both social facilitation and metabolic requirements (Ruckstuhl & Kokko, 2002).

Given this wide variety of models exploring collective behaviour, empirical tests exploring the individual behaviours driving observed collective behaviours are patchy in their coverage. Much research effort has been devoted to exploring how decision-making and leadership processes are connected and distributed within groups (Conradt & List, 2009; Dyer et al., 2009; King & Cowlishaw, 2009). Specific consideration of the effects of inter-neighbour interactions have explored individual decisions made during group movement according to metric (Herbert-Read et al., 2011; Ramseyer et al., 2009) or topological distance to neighbours (Ballerini et al., 2008; Nagy et al., 2010), and there have been a number of studies exploring leaving decisions (Sueur et al., 2011). Fewer studies have considered changes in behavioural state within a group. Several have considered how neighbours influence the vigilance patterns of groups (Beauchamp, 2009). Most tests of the models exploring changes in activity in response to metabolic requirements and the behaviour of neighbours (Ruckstuhl & Kokko, 2002) have focused on how differences in energetic requirements can lead to sexual segregation (Aivaz & Ruckstuhl, 2011; Michelena et al., 2008; Yearsley & Pérez-Barberia, 2005), non-synchronous behaviour (Šárová, Špinka & Panamá, 2007), group cohesion (Conradt, 1998), and group-size effects on activity (Gautrais et al., 2007). However, although these models assume that behavioural state is influenced by the actions of close neighbours, little has been done to test this empirically. Evidence is suggested by a study of cattle Bos taurus synchronising their lying behaviour, where their posture is more likely to be similar to neighbouring individuals compared to the rest of the herd (Stoye, Porter & Dawkins, 2012). However, there is scope for much more exploration of the assumptions behind models considering how the proximity of individuals to others can influence switches in their behavioural state. In this study, we asked whether the behaviours of individual red deer Cervus elaphus living in a managed herd are influenced by their neighbours. Individual deer spend large parts of their lives near or within large herds (Clutton-Brock & Albon, 1989), and therefore are ideal for addressing how changes in individual activity tie in with group-level behaviour. We hypothesised that deer that were topologically closer within the herd were more likely to be synchronised than would be expected when comparing two individuals randomly selected from different locations within the herd.

Materials and Methods

The work described is purely observational, conforming with UK law and ASAB/ABS guidelines on animal experimentation. Ethical approval was given by the University of Bristol Ethical Review Group (University Investigation Number UB/12/035).

The herd studied was housed in an enclosed 40.5 hectare deer park in the Ashton Court Estate, Bristol, England, composed of open grassland, with scattered patches of woodland. The herd is a population of c. 99 individuals of mixed age and sex, and its management and husbandry is conducted by Bristol City Council (the exact herd size was not known at the time of observation). Except for rutting periods, the enclosure is accessible to the general public, and the deer are habituated to the presence of humans and dogs. Permissions were not required for these observational studies, which occurred during the hours the public had access to the park. All observations were conducted within 10–100 m of the focal individuals, using binoculars where appropriate: for habituation, observers were in position for recording at least five minutes before observations started.

The study coincided with the rutting season of the deer, with stags often solitary and with greatly reduced feeding, and therefore likely to display very different behaviours to the rest of the herd (Clutton-Brock & Albon, 1989; Pépin, Morellat & Goulard, 2009). Males with antlers (approximately eleven individuals) were therefore excluded from the observations. The study focussed on females and young males that had not yet segregated from their maternal group, which were likely to display behaviour similar to the females (Clutton-Brock & Albon, 1989).

Prior to the study described, an ethogram was constructed for individual behaviour within the herd, differentiating between grazing, standing, walking, running, interacting, laying with head down, laying with head alert, and laying whilst ruminating. Within the analysis, these were reclassified as a combined dichotomous behaviours. Individuals were classified as ‘active’ if they were grazing, standing, walking, running, and interacting, and ‘resting’ if they were conducting one of the other behaviours.

For a single observation period, a focal individual was randomly selected from the herd. A random number between 1 and 99 was generated, and, considering the visible deer in the observer’s field of vision, the focal deer was selected by counting linearly from leftmost or rightmost visible deer (where the direction of counting was selected by a coin toss, and where a count was discarded if the random number selected was larger than the number of deer visible: this randomisation technique may have introduced some unavoidable bias towards individuals on the side of the herd closest to the observer, but, ignoring outlying stags, most of the herd was visible and countable during the sampling period and this bias should therefore have been minimal). Selected focal individuals were watched for twenty minutes. If the herd was disturbed by a human presence in the middle of the observation period, the observation was aborted and the data discarded. In total, eighteen complete observations of twenty minutes were conducted, over four days in October 2012; an additional two planned observations were started but aborted early due to disturbance, and have not been included in the analysis. All observations were conducted within 1200 and 1630 h, outside of the dawn and dusk peaks of activity frequently shown by red deer (Clutton-Brock, Guinness & Albon, 1982).

Over an observation period, the behaviour of the focal individual was recorded every minute. Simultaneously, the behaviour of the first, second and third closest individual in the herd to the focal were also recorded (where the identities of these individuals could change between the recording events as the deer moved within the herd). At the same time, the behaviour of a different randomly selected control individual within the herd was also recorded (selected using the randomisation technique described above from what remained of the herd after the focal and three nearest neighbours had been excluded, and ignoring rutting stags as stated above), where the identity of the control individual was independently chosen at each recording event.

Synchronisation between individuals was calculated as the proportion of the observations where the focal and test individual were both active or both inactive. Data did not follow the normality assumptions necessary for a repeated-measures analysis of variance, and were therefore compared with Friedman tests (Friedman, 1937). Because there was some chance that focal individuals were re-selected, there could be some degree of pseudoreplication in the dataset. To explore this, we generated a full set of Friedman tests where all possible combinations of up to five of the focal individuals were excluded from the analysis. Post-hoc tests were conducted for the comparison of synchronisation at different proximities, using two-tailed Wilcoxon signed-ranks tests assuming a normal approximation with continuity corrections, with the significance value adjusted to p = 0.009 using a Bonferroni correction. All analyses were conducted with R 3.0 (R Development Core Team, 2013). Raw data are presented in Supplemental Information 1.

Results and Discussion

Deer are less likely to be synchronised as they become socially further away from a focal individual (χ32=21.36, p < 0.001; Fig. 1). Because deer could not be individually identified, there is some chance that some pseudoreplication has occurred, with focal deer being resampled by chance. However, randomly removing data (all possible combinations of up to five focal individuals were removed) had no effect upon these results (the range of p values obtained fell in the range 0.000005–0.018). Post hoc tests demonstrated that focal individuals were more synchronised with first and second closest neighbours than with control deer (Fig. 1), but the increased synchronisation with the third-closest neighbour compared to the control (p = 0.011) was not significant after applying Bonferroni corrections.

Figure 1 Boxplot showing the proportion of time that behaviour of the focal individual was synchronised with neighbours of differing social distances.

Significant pairwise post-hoc tests are shown.

We used a dichotomous classification for behaviour, following the differentiation between ‘active’ and ‘passive’ behaviours used by Ruckstuhl & Kokko (2002). Individual deer were active for 73.11% of their time during the period observed (calculated by combining the individual datasets collected for focal, neighbour and control individuals: most of this active behaviour was grazing behaviour, as can be seen in Supplemental Information 2). If we assumed that all deer were acted independently of each other, then we can estimate that if we were to pick two individuals at random, they would be conducting the same action 60.68% of the time. This corresponds with the dotted line shown in Fig. 1, which falls near the middle of the control results. The three close neighbours were much more likely to be synchronised than this random estimate, suggesting that their individual behaviours are at least partially influenced by each other. The dichotomous scheme that we use may be falsely classifying some behaviours as similar (e.g., one member of an ‘active’ pair might be grazing whilst its partner is running). However, our dichotomous classification follows the differentiation between ‘active’ and ‘passive’ behaviours used by Ruckstuhl & Kokko (2002), which they demonstrate are sufficient to drive movement and segregation processes in ungulate-like animals. We would suggest that individuals conducting resting behaviours may have to invest more energy and expose themselves to a potentially greater risk of predation if they have to suddenly switch to one of the ‘active’ behaviours than if they were switching between two different ‘active’ behaviours or two different ‘resting’ behaviours. Therefore, considering just two behavioural states may be sufficient to try and pick apart broad patterns of synchronisation. Considering two easily-distinguished states also means that we are less likely to incorrectly classify finer-scale behaviours in the field that could look similar (such as the different ‘resting’ behaviours we initially recorded), although we do note that similar results are gained if we ignore this dichotomous classification and consider the exact synchronisation of the eight behaviour types recorded (Supplemental Information 3).

In considering the three nearest neighbours to a focal individual at a given moment in time, it was necessary to ignore a few factors which may have an effect on each individual’s behaviour. Firstly, the identity of each neighbour is likely to have changed over the course of consecutive observations of a focal deer. However, if we are interested in demonstrating that proximity is a factor driving behavioural synchronisation, this is not an issue as it is how the actions of the focal individual correlate with its unidentified neighbours that is important. Secondly, the observations do not account for inter-neighbour distance, where individuals in physically close proximity may be more likely to be synchronised. However, we are considering a topological relationship here (as is considered by Ballerini et al., 2008; Nagy et al., 2010) rather than a metric distance: it would be illuminating to observe whether increased physical proximity increases synchronisation, but the logistics of field observation made this too difficult to observe accurately. Thirdly, because this is an observational study, we are unable to separate whether synchronisation of activity is occurring in response to neighbour behaviour from whether some local effect is driving the behaviour instead: for example, deer that are close together may be more likely to be grazing because the quality of the local patch of grass available to them is better than that experienced by more distant individuals. Similarly, because we are looking at correlations, we are unable to separate mechanisms that may be causing local synchronisation from the observed behaviour: synchronisation could be occurring because key individuals are driving the local behaviours within the herd (King & Cowlishaw, 2009; Rands, 2011). To move from observing correlations to picking apart how synchronisation works, we would need to conduct experimental manipulations of the herd, such as by changing local forage quality or by removing possible key individuals from the herd.

The synchronisation behaviour we describe does not account for social relationships between the individuals. Local social networks are likely to strongly influence substructures within groups (Bode, Wood & Franks, 2011b; Sueur et al., 2011), and being able to identify individuals and assay their interaction behaviour over longer periods of time may give us a much clearer indication of the behavioural dynamics of the herd. Similarly, we did not account for how differences in the physiological state (Rands et al., 2003; Rands et al., 2008; Rands et al., 2006) or social status (Rands, 2011) of individuals could be influencing their need to copy the behaviour of others. There is still a need to properly link models and empirical work considering how social foraging behaviour can influence group behaviour (Marshall et al., 2012), and in particular we urge further studies of the effects of neighbour proximity in order to explore these neglected assumptions implicit within many models.

This study demonstrates that individual deer are more likely to synchronise their activity with their closer neighbours when compared to more distant neighbours and the wider herd. This provides support for the spatial assumptions used in models of activity synchronisation (Ruckstuhl & Kokko, 2002). Similar patterns were seen in small herds of cattle (Stoye, Porter & Dawkins, 2012), but the current study demonstrates that these assumptions may also be applicable to much larger herds of animals.

Supplemental Information

Supplemental Information 1 Raw behavioural data

Data presented as a comma-delimited file: full details of how the data is encoded is given at the beginning of the file.

Click here for additional data file.

Supplemental Information 2 Figure showing overall proportion of observations where deer were conducting each of the classified behaviours

Click here for additional data file.

Supplemental Information 3 Boxplot showing the proportion of time that behaviour of the focal individual was synchronised with neighbours of differing social distances, considering the exact behaviour the focal is conducting

Synchronisation of individual behaviours between the focal individual and its neighbours or a control individual was assessed by calculating the proportion of observation periods that each focal individual was conducting exactly the same behaviour as the compared individual, considering each of the eight possible behavioural classes recorded as a different behaviour. Data were compared using a Friedman test, as they did not fit assumptions of normality, and post hoc analyses were conducted using two-tailed Wilcoxon signed-ranks tests assuming a normal approximation with continuity corrections, with the significance value adjusted to p = 0.009 using a Bonferroni correction. Deer at different social distances differed in their level of synchronisation with the focal individual (χ32=19.29, p < 0.001), and pairwise comparisons demonstrated that the focal individuals were more likely to be synchronised with their closest neighbour than with a control individual (p = 0.001); all other pairwise comparisons were non-significant.

Click here for additional data file.

Andrew Robins, Benjamin Dalziel, and four anonymous referees who commented on earlier versions of this manuscript are thanked.

Additional Information and Declarations

Competing Interests

Author Contributions

Animal Ethics

The authors declare there are no competing interests.

Sean A. Rands conceived and designed the experiments, analyzed the data, wrote the paper, prepared figures and/or tables, reviewed drafts of the paper.

Hayley Muir and Naomi L. Terry conceived and designed the experiments, performed the experiments, analyzed the data, reviewed drafts of the paper.

The following information was supplied relating to ethical approvals (i.e., approving body and any reference numbers):

The work described is purely observational, conforming with UK law and ASAB/ABS guidelines on animal experimentation. Ethical approval was given by the University of Bristol Ethical Review Group (University Investigation Number UB/12/035).

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
