# Peer review of "Red deer synchronise their activity with close neighbours"

_PeerJ, doi:10.7717/peerj.344_

## Round 0.1 · original submission · Major Revisions

The reviewers had differing opinions on your paper. My own reading of it, leads me to suggest that you should address the concrete points raised by the reviewers and write an explanation addressing all of the reviewers' comments and explaining the changes you have made to the manuscript.

Reviewer 1 ·

Basic reporting

"No Comments"

Experimental design

The main goal of this paper is very interesting and valuable. However, because of methodological problems the results presented here should be questioned. Much information is missing and the observation period is simply too short to meet the objectives of the paper.

Validity of the findings

The manuscript is not very carefully prepared, and the analyses and discussion are not strong.

Comments for the author

I encourage the authors to design fresh studies to investigate this topic using a more accurate and extended time sampling. Strong studies on this topic will be of considerable interest to the PeerJ audience

·

Basic reporting

I found the paper very well presented and clearly written, with only a few minor problems with organisation in the Materials/Methods and Results/Discussion sections. Specifically, I feel that the statements of line 120-122 (Ruckstuhl & Kokko 2002) would be better positioned around line 80 where behavioural dichotomy was first mentioned.

Line 43: "(Šárová et al. 2007)" not hyperlinked with bibliography
Line 99: "even." should be replaced with "event"
Line 123: Please be more specific for the statement "considering the entire dataset..." - perhaps remind the reader that this pertains to all separate scores from focal, two nearest-neighbour and control animals.

Experimental design

The selection of the focal animal for each scored observation was well conceived and explained, however I am left with some doubt as to the selection of the "control" animal. I am left with the impression that there may have been unconscious bias in its selection - perhaps the control was selected as it was most visible, thus being more peripheral to the herd, thus (conceivably) engaged in more vigilant/active sets of behaviours. It is not probably possible to gauge the centrality in the herd group of the focal/control animal as no photography or filming took place, but the lack of detail in the selection of the control is the only aspect that weakens the presented argument - but this is not a fatal flaw.
The control animal was also labelled "random" in Figure 1 - perhaps one label be consistently used to avoid confusion in the reader.
Some detail was omitted - how many discarded antlered males were in the herd group, to provide a more precise number of females/young in the observation group rather than "c. 99" (Line 63).
How often was the herd disturbed - there is mention that some 20-min observation periods were discarded due to disruption by humans (lines 88-89), but it would be more helpful to know what percentage of observation periods were aborted.

Validity of the findings

Validity of the findings is good such that the methods can be replicated by others and collective behaviour models further tested with photography/filming/proximity analyses - as the authors discuss.

Comments for the author

The points and queries I raised in the previous comments are provided only for the author's guidance should they wish to further improve the paper. I am satisfied that the paper could be otherwise accepted as presented.

---

## Round 0.2 · Major Revisions

Since reviewers 1 and 2 had completely different views about the publication of your manuscript, I sent your revised manuscript to a third reviewer, who has offered some constructive suggestions. Please take all of these into account in another revision of your paper.

·

Basic reporting

I think the literature review has a good coverage, but could be stated more succinctly, especially given the simplicity of the experiment and findings. There are statements in the lit review and the intro in general that could be made more specific. Please see general comments.

Experimental design

I think the analytical approach needs to be more carefully explained/justified, and more analyses (of the finer scale behavior data) may be necessary. Please see general comments.

Validity of the findings

The boxplot in Figure 1 is convincing, but the explanation of the analysis needs some work. In particular, the authors should also analyze the more fine-scale behavioral data they collected (not just the dichotomous data), or carefully explain why this analysis is not included.

Comments for the author

Review of Rands et al. submitted to PeerJ Jan 2014

This paper describes an observational study in which the behaviors of nearby Red Deer (Cervus elaphus) were tested to see if they were more similar than expected for independent individuals. The authors show that neighbors tend to be in the same dichotomous activity state (active versus non-active) more often than expected by chance.

The main result is evident in the boxplot of Figure 1, but I think there are some places where the explanation of the analytical approach could be improved.

My main concern is that the behavioral data were collected on a finer scale – including states like grazing, walking, lying down – and then analyzed at a courser scale – classifying each behavior as either active or passive.

While I understand the potential advantages of the dichotomous approach, I do wonder what the analysis reveals if it is done on the finer scale data. It could be really interesting... e.g. are some behaviors more likely to be correlated among neighbours than others? I suggest the authors include a finer scale analysis as well (even if it produces no statistically significant differences, that is not a reason to not show it – on the contrary), or more convincingly explain why it is omitted.

The first paragraph of introduction might benefit from specific examples. What exactly are some of the benefits of grouping? What aspects of collective behavior observed in natural systems are simulations good at reproducing? And how does the ‘noisiness’ of biological systems (incidentally, what is meant by noise – observation error, or?) increase difficulty in studying interaction rules?

The literature review has a good coverage, but I think it is written a little long for the simple experiment it prefaces. There is more here than is needed to put the main result in the proper context, and some of it is unfocussed. Would it be possible to keep the breadth, but be a little more succinct, and specific?

Might want to take a look at:
Haydon, D. T., Morales, J. M., Yott, A., Jenkins, D. A., Rosatte, R., & Fryxell, J. M. (2008). Socially informed random walks: incorporating group dynamics into models of population spread and growth. Proceedings of the Royal Society B: Biological Sciences, 275(1638), 1101–1109.

Line 84 – What is a two figure random number?

Line 109 - Freidman test should have a reference and brief explanation.

Line 110 - On ‘pseudoreplication’ – I think the authors should explain in more detail how reselecting the same focal individual might affect the results.

I personally (others legitimately differ) have a negative reaction to the term pseudoreplication, unless it is carefully qualified. I think pseudoreplication masquerades as a technical statistical term that implies your data cannot be useful. In fact the term was invented by ecologists
in the context of a particular (and rather ancient) set of experimental and analytical approaches upon which we have become accustomed to relying. Many datasets have complex correlation structures and it is not necessarily a bad thing – although it may necessitate more detailed or focused analyses than can be achieved with ANOVA-like approaches.

Line 123 and throughout - replace ‘post hoc’ with the specific test used, or omit.

The distributions shown by the box plot are the main thing that convinces me. I wonder if a statistical test comparing distributions might be another avenue to pursue for the analysis.

Line 128 – unclear how the dichotomous classification system is particularly related to the synchronization by chance, which can happen for many measures of synchrony. Are you saying that as you reduce the number of categories the probability of two individuals being in the same category by chance increases? And why only use two categories when more fine scale behavioral data was collected? What are the results when you repeat the analysis with the fine scale data?

In results/discussion, the tone seems defensive to me as a statement of a result is often followed by two or more lines of caveats and acknowledgements. It might improve the ‘feel’ of this section to aggregate the results you can state confidently and begin the section with that (e.g. nearest neighbors had a significantly higher probability of being in the same state than expected by chance). Then later put a paragraph that argues for the (limited) inferential scope of the results to broader questions of collective behavior.

Line 138 – ‘However, our dichotomous classification follows the differentiation between ‘active’ and ‘passive’ behaviours used by Ruckstuhl & Kokko (2002).’ This isn’t much of a reason per se. What did Ruckstuhl & Kokko write in 2002 that justifies using a dichotomous approach?

Line 174- Try to avoid emphatic adjectives like ‘great’ (as in ‘great need’) unless they significantly alter the meaning of a sentence. Same with ‘careful studies.’

Throughout – consider the effect of omitting the phrase ‘we acknowledge’ in each sentence where it appears. Does it affect the meaning of the sentence? If it doesn’t, remove it.

---

## Round 0.3 · accepted · Accept

Thank you for your input and patience.

·

Basic reporting

No Comments

Experimental design

No Comments

Validity of the findings

No Comments

Comments for the author

The latest revision of the manuscript addresses the most substantial points I raised regarding the analysis and presentation of the results.